

# Shared genomic outliers across two divergent population clusters of a highly threatened seagrass

Nikki Leanne Phair[1], Robert John Toonen[2], Ingrid Knapp[2] and Sophie von der Heyden[1]

[1] Department of Botany and Zoology, University of Stellenbosch, Stellenbosch, South Africa
[2] Hawaii Institute of Marine Biology, University of Hawaii at Manoa, Kaneohe, Hawai'i, United States of America

## ABSTRACT

The seagrass, *Zostera capensis*, occurs across a broad stretch of coastline and wide environmental gradients in estuaries and sheltered bays in southern and eastern Africa. Throughout its distribution, habitats are highly threatened and poorly protected, increasing the urgency of assessing the genomic variability of this keystone species. A pooled genomic approach was employed to obtain SNP data and examine neutral genomic variation and to identify potential outlier loci to assess differentiation across 12 populations across the ~9,600 km distribution of *Z. capensis*. Results indicate high clonality and low genomic diversity within meadows, which combined with poor protection throughout its range, increases the vulnerability of this seagrass to further declines or local extinction. Shared variation at outlier loci potentially indicates local adaptation to temperature and precipitation gradients, with Isolation-by-Environment significantly contributing towards shaping spatial variation in *Z. capensis*. Our results indicate the presence of two population clusters, broadly corresponding to populations on the west and east coasts, with the two lineages shaped only by frequency differences of outlier loci. Notably, ensemble modelling of suitable seagrass habitat provides evidence that the clusters are linked to historical climate refugia around the Last Glacial Maximum. Our work suggests a complex evolutionary history of *Z. capensis* in southern and eastern Africa that will require more effective protection in order to safeguard this important ecosystem engineer into the future.

## INTRODUCTION

Despite potentially high levels of gene flow, signals of local adaptation to environmental factors such as salinity and temperature gradients, have been described for a diverse set of marine species (*Guo et al., 2015*; *Guo, Li & Merila, 2016*; *Dalongeville et al., 2018*; *Nielsen et al., 2018*), and to osmotic niches in freshwater species (*Dennenmoser et al., 2016*; *Attard et al., 2018*; *Lucek et al., 2018*). These and other studies suggest that contemporary spatial patterns of outlier loci significantly contribute towards shaping genetically structured populations (*Savolainen, Lascoux & Merilä, 2013*; *Yeaman, 2013*; *Huang et al., 2014*;

Corresponding author
Sophie
von der Heyden, svdh@sun.ac.za

*Tigano & Friesen, 2016*; *Barth et al., 2017*; *Cure et al., 2017*; *Marques, 2017*), although their relevance to local adaptation is often unknown. While standing genomic variation is the material on which selection can act, adaptive variation is expected to increase evolutionary resilience by improving the ability to persist through and adapt to changing environmental conditions (*Bible & Sanford, 2016*). However, in addition to present-day environmental conditions, historical processes should also be considered, as they often play an important role in shaping contemporary patterns of genomic diversity and differentiation (*Hewitt, 2000*; *Gaither et al., 2015*; *Toms et al., 2014*; *Leprieur et al., 2016*; *Chefaoui, Duarte & Serrão, 2017*; *Hernawan et al., 2017*), that could impact the distribution and maintenance of contemporary patterns of neutral and potentially adaptive variation. If the latter is linked to gene regions of known function, this may signal some adaptive importance (*Angeloni et al., 2012*; *Hoban et al., 2016*) and can better our understanding of the mechanisms behind adaptation.

RADSeq (Restriction Site Associated DNA Sequencing) methods have been used to investigate outlier loci and have identified both directional (*Hohenlohe et al., 2010*; *Lexer, Wüest & Mangili, 2014*; *Gaither et al., 2015*) and stabilising selection patterns consistent with adaptation in several studies (*Hohenlohe et al., 2010*; *Gaither et al., 2015*), providing unique insights into the evolutionary mechanisms of non-model species. However, our understanding of how spatial variation of outlier loci among populations might contribute towards shaping population divergence in natural systems can still be further developed. In addition, it can be challenging to disentangle the signatures of potential adaptation to different environments from the simple isolating effect of distance, especially as a high degree of collinearity exists between environmental gradients and neutral population structure (*Wang & Bradburd, 2014*; *Manthey & Moyle, 2015*; *Prunier et al., 2017*; *Weber et al., 2017*; *Rodríguez-Zárate et al., 2018*). In broad spatial and environmental contexts, both Isolation By Distance (IBD) and Isolation By Environmental (IBE) will act in differentiating populations. While patterns of IBD have been observed in organisms across a range of life histories and taxonomic groups (*Kelly, MacIsaac & Heath, 2006*; *Van Dijk et al., 2009*; *Harris & Taylor, 2010*; *Moura et al., 2014*; *Wright et al., 2015*), the contribution of IBE in marine systems is becoming more apparent (*Limborg et al., 2009*; *Mendez et al., 2010*; *Whittaker & Rynearson, 2017*).

Within this context, determining the spatio-temporal patterns of genomic variability that may provide some insights into signals of adaptation of populations, is important for understanding persistence and resilience of species (*Sexton, Hangartner & Hoffmann, 2014*; *Bernatchez, 2016*), especially those under threat from environmental pressures. Importantly, detecting potentially adaptive variation can assist in pinpointing conservation units, as local adaptation is an important part of evolutionary diversification, even on a contemporary timescale (*Bible & Sanford, 2016*; *Bonin et al., 2007*; *Carvalho et al., 2011*; *Funk et al., 2012*; *Hanson et al., 2017*; *Von der Heyden, 2017*). Globally, seagrasses are facing persistent declines and habitat fragmentation (*Orth et al., 2006*; *Waycott et al., 2009*), both of which have been linked to loss of genetic diversity (*Orth et al., 2006*; *Williams, 2017*). Decreased population sizes and loss of genetic diversity are particularly important in the

face of climate change and habitat alteration facing coastal systems such as the habitat of the southern and eastern African seagrass, *Zostera capensis* (Setchell; family Zosteraceae).

*Zostera capensis* has a disjunct distribution limited to estuaries on the cool-temperate biogeographic region on the west coast, the warm-temperate south coast and the sub-tropical east coast of South Africa, where it is the dominant seagrass, and extends up the tropical east African coast to sheltered bays in Kenya. The wide distribution range of this vulnerable species (IUCN; *Short et al., 2010*; *Green & Short, 2003*; Fig. 1A) encompasses strong environmental gradients across multiple biogeographic regions providing an excellent opportunity to study the genomic variation of relatively isolated populations along a wide gradient of environmental conditions. *Zostera capensis* likely relies largely on vegetative reproduction (*Greve & Binzer, 2004*; *Hall, Hanisak & Virnstein, 2006*), as flowering in this species has only been recorded once under specific laboratory conditions (*McMillan, 1980*). Unfortunately, the dispersal potential of vegetative fragments is unlikely to provide meaningful connectivity between sites due to harsh coastal conditions, strong currents and often long distances between suitable estuarine habitats (*Weatherall et al., 2016*).

Previous studies have shown that the genetic diversity, clonality and connectivity of seagrasses globally is highly context dependent (*Jover et al., 2003*; *Olsen et al., 2004*; *Procaccini, Olsen & Reusch, 2007*; *Sinclair et al., 2014*; *Arriesgado et al., 2016*; *Kendrick et al., 2016*; *Hernawan et al., 2017*; *Putra et al., 2018*), with some studies reporting high genetic diversity and population structuring at regional and local scales (*Diekmann et al., 2005*; *Van Dijk & Van Tussenbroek, 2010*; *Becheler et al., 2010*; *Sherman et al., 2016*), emphasizing the role of near and off-shore currents (*Muñiz Salazar et al., 2005*; *Nakajima et al., 2014*). Conversely, in a few cases, low levels of genetic diversity and shared genotypes, even across exceptionally large spatial scales, have been recorded (*Van Dijk & Van Tussenbroek, 2010*; *Evans et al., 2014*; *Nakajima et al., 2014*; *Phan et al., 2017*). So-called 'mega clones' of *Thalassia testudinum* can even have single ramets dispersed over 47 km (*Bricker et al., 2018*) and 'millenary clones' of *Posidonia oceanica* are estimated to be hundreds to thousands of years old (*Arnaud-Haond et al., 2012*; *Ruggiero, Turk & Procaccini, 2002*). However, how patterns of natural variation and population genomic structure in seagrass are shaped by adaptive processes remains poorly explored.

Changing African seascapes, through anthropogenic and changing climate pressures (*Mead et al., 2013*), are severely impacting local populations of *Z. capensis*, prompting calls to monitor and map genomic variation, both for neutral and outlier loci that may indicate some adaptive variation. We utilised a pooled RADseq approach to identify patterns of variation in both neutral and outlier loci for populations throughout the total range of *Z. capensis*, with the underlying hypothesis that signals of outliers would vary among populations, given that each site experiences a unique combination of environmental conditions. We also examined the predicted historical distribution by means of hindcast species distribution modelling, as historical conditions are likely to have a strong influence on contemporary patterns of diversity in southern Africa (*Toms et al., 2014*). Lastly, we examine the role of geographical and ecological distance in shaping patterns of variation of *Z. capensis* in its environmentally heterogeneous habitat and hypothesise that IBE will be at least as important as IBD in driving genomic diversity.

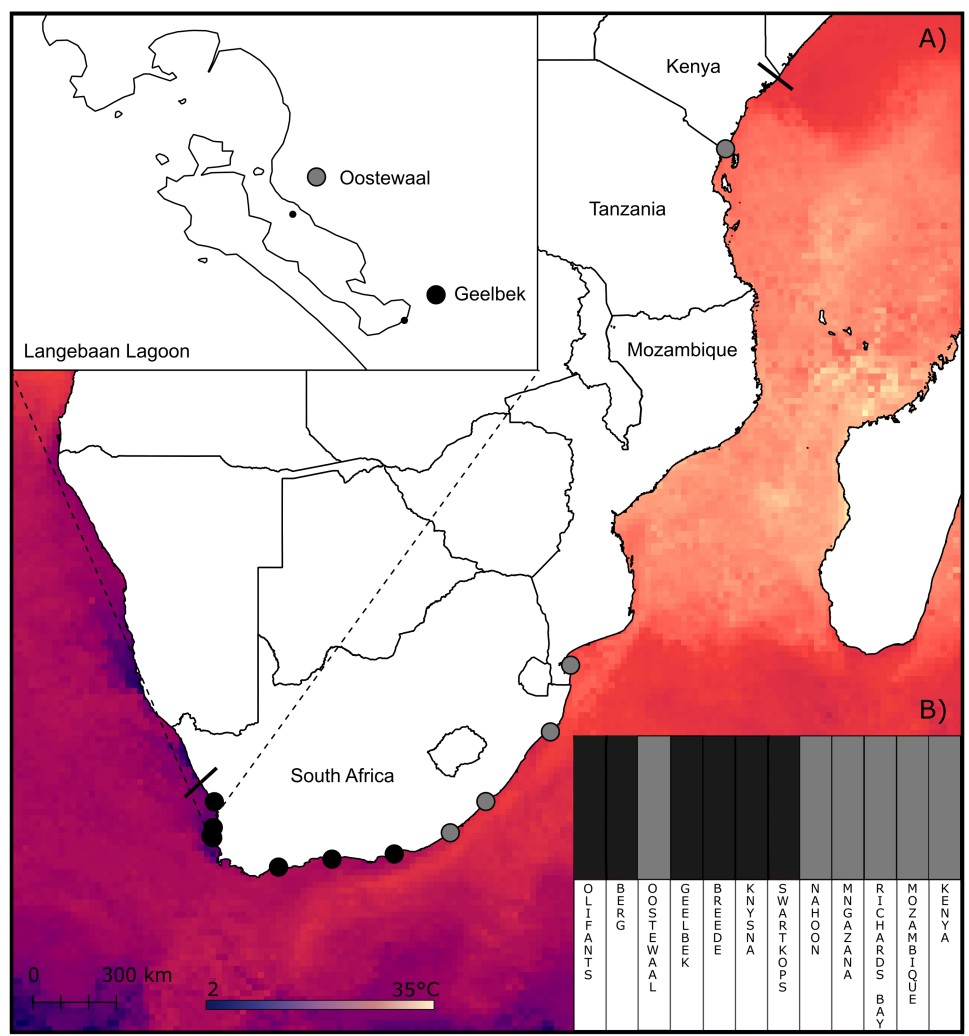

**Figure 1** **Sampling sites and clustering of *Z. capensis* populations.** (A) Map showing the sampling sites and sea surface temperature across the range of *Z. capensis* (indicated by the solid lines). An inset of the two sites at Langebaan Lagoon is provided. (B) Clustering analysis of the twelve sites estimated in BAPS for the complete dataset, with the twelve sites grouped into two clusters.

## MATERIALS AND METHODS

### Sample collection

Leaf samples ($n = 336$) were collected from 12 sites, including nine estuaries/estuarine bays along the South African coast, one bay in Mozambique (Inhaca) and one bay in Kenya (Shimoni; Fig. 1A; Table 1). At each location, with the exception of Inhaca, and Shimoni, three leaf samples were collected 2 m apart from five beds at two sub-sites (>100 m) for a total of 30 leaf samples per location. This sampling approach was designed to minimise the sampling of clones. Samples were preserved with silica crystals before being processed. Sampling permits were granted by SanParks and CapeNature (permit number 0028-AAA008-00159); DAFF and DEA (permit number was RES2014/103).

**Table 1** Sampling locations, biogeographic zone and number of samples (*N*) per site.

| Location | Abbreviation | Coordinates | | Biogeographic zone | N |
|---|---|---|---|---|---|
| Olifants | O | 31.7021°S | 18.1876°E | Cool-temperate Namaqua | 30 |
| Berg | B | 32.7697°S | 18.1438°E | Cool-temperate Namaqua | 30 |
| Geelbek, Langebaan | L1 | 33.1941°S | 18.1211°E | Cool-temperate Namaqua | 30 |
| Oostewaal, Langebaan | L2 | 33.1214°S | 18.0447°E | Cool-temperate Namaqua | 30 |
| Breede | BR | 34.4074°S | 20.8453°E | Warm-temperate Agulhas | 30 |
| Knysna | K | 34.0791°S | 23.0562°E | Warm-temperate Agulhas | 30 |
| Swartkops | SK | 33.8650°S | 25.6333°E | Warm-temperate Agulhas | 30 |
| Nahoon | N | 32.9864°S | 27.9517°E | Warm-temperate Agulhas | 30 |
| Mngazana | M | 31.6921°S | 29.4228°E | Suptropical Natal | 30 |
| Richards Bay | RB | 28.8105°S | 32.0947°E | Suptropical Natal | 23 |
| Inhaca, Mozambique | MOZ | 26.0500°S | 32.9297°E | Tropical Delagoa | 10 |
| Shimoni, Kenya | KEN | 4.6741°S | 39.3440°E | Tropical | 3 |

Despite intensive questioning of collaborators and other contacts throughout this study, no samples of *Z. capensis* were obtained from Tanzania (L Nordlund, pers. comm., 2015).

## Laboratory protocols

Accurately estimating genome-wide variation and detecting signals of local adaptation in non-model organisms, such as seagrasses, requires many individuals from many sites to be sequenced, which can be prohibitively expensive despite the advances made by high-throughput sequencing methods such as RADseq (*Ellegren, 2014*; *Andrews et al., 2016*). As such, a pooled sequencing (pool-seq) approach was utilised, combining the genomic DNA of multiple individuals before sequencing (*Sham et al., 2002*; *Kofler, Betancourt & Schlötterer, 2012*). This approach decreases cost whilst increasing the number of individuals analysed and allowing for a more population focussed analysis (*Futschik & Schlötterer, 2010*; *Schlötterer et al., 2014*).

Genomic DNA was extracted from leaf tissue using Qiagen DNeasy plant kit (Qiagen, Valencia, USA) following standard protocols, with the exception of eluting the DNA in nuclease-free water instead of elution buffer. Genomic DNA quality was then assessed using gel electrophoresis and DNA concentrations of each sample determined by Qubit analysis at the Central Analytical Facility of Stellenbosch University (CAF). Genomic DNA was pooled by location, with all individuals having equimolar representation, to create a total of 12 samples for Illumina sequencing. The two sites at Langebaan, Oostewaal and Geelbek (Fig. 1A), were kept as separate pools to allow for comparison between the observed morphotypes; one short and stunted on the muddy tidal flats (Geelbek) which experience prolonged exposure to conditions outside the water and the other is longer with a higher biomass on the sandy permanently submerged area (Oostewaal) (D Pillay, pers. comm., 2014).

Library preparation and sequencing followed the ezRAD method (*Knapp et al., 2016*; *Toonen et al., 2013*; *Nielsen et al., 2018*), which obtains a reduced representation sequencing library using high frequency restriction enzymes. Pooled genomic DNA was freeze dried

before library construction following the protocol of *Knapp et al. (2016)* and Mi-Seq Illumina sequencing (V3 2x300) conducted at the Genetics Core Facility (GCF) of the Hawaii Institute of Marine Biology (HIMB). These data are stored in the National Center for Biotechnology Information's (NCBI's) Sequence Read Archive (SRA; PRJNA503110) and georeferenced at GeOMe (https://geome-db.org) at the project: *Zostera capensis* pooled RADseq.

## Data processing and alignment

Quality of raw reads was analysed using FastQC (*Andrews, 2010*) and quality filtering carried out using FastQC Toolkit (*Andrews, 2010*), removing low quality bases (<20 phred score). Additionally, TrimGalore! v 0.4.4 (available at: http://www.bioinformatics.babraham.ac.uk/projects/trim_galore/) was used to remove any remaining adapter sequences or reads shorter than 30 bp. BWA-MEM (*Li, 2013*) was used to map filtered paired-end reads from each pooled sampling site to the genome of sister species, *Zostera marina* (available from NCBI, BioProject number PRJNA41721, GenBank accession number LFYR00000000 (*Olsen et al., 2016*), with a minimum mapping quality of 20. Ambiguously mapped reads, PCR duplicate reads, reads with less than 20 mapping quality and less than 20 base quality were filtered out before converting SAM files to BAM files with SAMtools (*Li et al., 2009*). Number of mapped and unmapped reads were then calculated using the *idxstats* command in SAMtools. As pools with a higher number of mapped reads may have an artificially inflated number of SNPs, mapped reads were subsampled to median coverage in SAMTools using the *view* command with the '-s' flag. Although subsampling results in a loss of data, it is nonetheless important for correctly interpreting true differences between the sampling sites, as opposed to differences in data quality or quantity (*Schlötterer et al., 2014*). To confirm that subsampling removed any possible correlation between the number of mapped reads and the number of SNPs and outlier loci identified downstream, Spearman's correlation coefficients were calculated using the *rcorr* function of the 'Hmisc' (*Harrell Jr & Dupont, 2006*) package in R (*R Core Development Team, 2008*). BAM files were sorted and indexed before being used to create pileup files for each individual sampling site with the *mpileup* command in SAMTools (*Li et al., 2009*), using a minimum quality score of 20 and maximum read depth of 10,000. Finally, a pileup file combining all sites was created using the same parameters in SAMtools and converted to a sync file using PoPoolation2 (*Kofler, Pandey & Schlötterer, 2011*) for downstream use.

## Calling SNPs and simulating data

The total number of SNPs and private SNPs were identified using *snp-frequency-diff.pl* in PoPoolation2 (*Kofler, Pandey & Schlötterer, 2011*) with genomic sites required to have a minimum minor allele count of four, and coverage between 10 and 500 across all 12 sampling sites. SNPs were then filtered to retain only those present among sampling sites, and not those present due to differences between the reference sequence (*Z. marina*) and *Z. capensis*. As many software cannot handle pooled data, requiring individuals to be specified within sampling sites, *subsample_sync2GenePop.pl* in PoPoolation2 was used to simulate a multi-locus dataset of a subset of SNPs identified by PoPoolation2. Because this programme

cannot simulate different numbers of individuals across sites, the median sample size of 30 individuals was selected for every site. The resulting GenePop file was then converted to various formats in PGDspider (*Lischer & Excoffier, 2012*) for downstream analyses.

## Outlier loci identification and functional annotation

Due to the uncertainty surrounding RADseq, Pool-seq and outlier detection methods (*Narum & Hess, 2011*; *Da Fonseca et al., 2016*; *Mckinney et al., 2016*; *Lowry et al., 2017*; *O'Leary et al., 2018*), four outlier detection methods were employed, namely BayeScan v2.1 (*Foll & Gaggiotti, 2008*), Lositan (*Antao et al., 2008*), BayeScEnv (*De Villemereuil & Gaggiotti, 2015*) and PCadapt (*Luu, Bazin & Blum, 2016*), which includes $F_{ST}$-based approaches, genotype-environment correlations and principle component analyses (see supplementary materials for details).

Outlier loci identified by two or more methods were considered "candidate outliers" and their functional roles evaluated by subjecting 1,000 base pairs upstream and downstream of each of the identified outlier SNPs to BLASTx searches, with the non-redundant protein sequence database and an *E*-value cut off of $10^{-5}$ (*Altschul et al., 1997*) using Blast2Go (*Conesa et al., 2005*). In addition to BLASTing against the general NCBI database, these searches were also carried out against the Zosteraceae family in general, specifically, *Zostera marina* (*Olsen et al., 2016*) and *Z. muelleri* (*Lee et al., 2016*). Gene Ontology (GO) mapping, Interproscan (*Jones et al., 2014*) and annotation were performed with Blast2Go default parameters. The number and proportion of candidate outliers unique to each site and shared between pairwise sites was calculated. The overlap of outliers identified between the different approaches was visualised using the 'VennDiagram' package (*Chen & Boutros, 2011*) in R. The frequency of outlier loci identified by all four approaches (Lositan, BayeScan, BayeScEnv and PCAdapt) was plotted across sampling sites using the 'ggplot2' package (*Wickham, 2009*) in R and listed in Table S3. A one-way analysis of variance (ANOVA) and a post hoc TukeyHSD test, were performed with the 'aov' and 'TukeyHSD' functions in R, to compare outlier frequencies across sites.

## Neutral variation

All identified outlier loci were flagged as being putatively under selection for the purposes of this analysis and were therefore removed from the dataset to isolate neutral drivers of patterns of population structure. The neutral-only multi-locus dataset set was then re-simulated, using *subsample_sync2GenePop.pl* in PoPoolation2 with 30 individuals per site as described above, and used to calculate measures of neutral variation.

## Genome-wide variation and differentiation

To characterise genetic diversity, Tajima's nucleotide diversity ($\pi$), Watterson's theta ($\theta$) and Tajima's D were estimated for the complete and neutral-only datasets using a sliding window approach with *Variance-sliding.pl* in PoPoolation v1.2.2 (*Kofler et al., 2011*). For these comparisons, filtering stringency was reduced to a minimum minor allele count of two and coverage between 10 and 500 per sampling site. As the estimation of allele frequencies in pooled individuals is highly reliant on sequence coverage, a high sequence coverage and large sliding windows were used in order to increase accuracy (*Kofler et al.,*

*2011*). Observed and expected heterozygosity and the inbreeding coefficient ($F_{IS}$) was estimated from the simulated datasets with the *divBasic* function of the 'DiveRsity' package (*Keenan et al., 2013*) in R.

To investigate genome-wide levels of differentiation, the fixation index ($F_{ST}$) for pairwise comparisons of populations was estimated using a sliding window approach with *fst-sliding.pl* in PoPoolation2, using a minimum minor allele count of four and a coverage between 10 and 500. Fisher's exact test was carried out with *fisher-test.pl* in PoPoolation2 to estimate the significance of allele frequency differences between sites. Patterns of differentiation were visualised on a principle coordinates analysis (PCoA) plot generated in R using the *pco* function of the 'labdsv' package (*Roberts, 2007*). The PCoA plot was generated both with and without Kenya in order to account for sampling bias. The simulated neutral dataset was used to investigate population clustering by means of Bayesian Analysis of Population Structure (BAPS) software (*Corander & Marttinen, 2006*; *Corander, Marttinen & Mäntyniemi, 2006*) testing $K = 1–10$.

## Habitat suitability for *Z. capensis* in the LGM

In order to understand the influence of historical environmental conditions on the contemporary patterns of genomic variability, the suitable habitat for *Z. capensis* was hindcast to the Last Glacial Maximum (LGM; 21 kya). *Zostera capensis* occurrence data was obtained from *Adams, Veldkornet & Tabot (2016)* and environmental data downloaded from MARSPEC at 5 arcminute resolution for both the present-day (*Sbrocco & Barber, 2013*) and the LGM (CNRM-CM33 model; *Braconnot et al., 2007*; *Sbrocco, 2014*). Following *Chefaoui, Duarte & Serrão (2017)*, only Sea Surface Temperature (SST) of the coldest month (Biogeo14) and warmest month (Biogeo15) were utilised to avoid using strongly correlated variables for Species Distribution Modelling (SDM; *Guisan & Thuiller, 2005*; *Braunisch et al., 2013*; *Chefaoui, Duarte & Serrão, 2017*). Precipitation variables were excluded to decrease model uncertainty (*Varela, Lima-Ribeiro & Terribile, 2015*). These variables represent relevant present-day and LGM conditions, which are recognised as important determinants of intertidal seagrass habitat suitability (Short Neckles, 1999; *Short et al., 2010*; *Valle et al., 2014*) and they are projected along the present-day (*Sbrocco & Barber, 2013*) and LGM coastlines (*Braconnot et al., 2007*; *Sbrocco, 2014*), respectively. Environmental variables such as salinity and oxygen content were not included due to the high natural variability in estuarine systems over tidal and seasonal time-scales. QGIS (*QGIS Development Team, 2012*) was used to crop raster extents, by means of the buffer and crop tools, to focus on the coastal areas including and surrounding the present-day range of *Z. capensis*.

Ecological niche modelling was implemented through an ensemble approach with the 'biomod2' package (*Thuiller et al., 2016*) in R. As in *Chefaoui, Duarte & Serrão (2017)*, the following six presence-absence algorithms were included in the ensemble models: generalized additive model (GAM), flexible discriminant analysis (FDA), generalized boosting model (GBM), multiple adaptive regression splines (MARS), generalized linear model (GLM), and random forest (RF). Default parameters were used for all algorithms, except for the GLM which was fitted with a quadratic term, the GBM which was run

with 1,000 trees, and the GAM which was executed with the GAM_mgcv function. As the occurrence data (*Adams, Veldkornet & Tabot, 2016*) included reliable presence and absence records for estuaries along the entire South African coastline, no pseudo-absence selection was required. The data was split into a calibration (80%) and a validation (20%) set and three iterations were performed for each algorithm with three permutations to estimate and weight variable importance, for a total of 18 models. Models were assessed with the true skill statistic (TSS; *Allouche, Tsoar & Kadmon, 2006*) and the area under the receiver operating characteristic (ROC) curve (AUC; *Fielding & Bell, 1997*), considering both specificity (true negatives) and sensitivity (true positives). Only models scoring TSS >0.55 and AUC >0.8 were used to produce ensembles. Retained models were ensembled to produce a weighted mean SDM and first used to project the present-day habitat suitability, in terms of SST, along the South African coastline, and then used to hindcast the habitat suitability to the LGM. The present-day and LGM habitat suitability projections, as well as the changes in habitat suitability between the present-day and LGM were plotted in R.

### Disentangling contemporary signals of IBD and IBE

A redundancy analysis (RDA) (*Legendre & Legendre, 2012*) was conducted to evaluate the relative contribution of spatial and environmental variation to genomic variability and patterns of genetic differentiation. RDA can be useful as a multivariate regression technique when running regression analyses with multivariate predictors (space and environment) and multivariate responses (here, minor allele frequencies of SNPs). As spatial distances are not suitable for constrained ordination or regression as implemented in RDA, geographic distances were transformed to Principal Coordinates of Neighbourhood Matrix (PCNM) distances with the *pcnm* function in the 'vegan' package (*Oksanen et al., 2015*) in R. Environmental distances were calculated within the RDA function from the variables in Table 2 (excluding the macrophyte species measure, which were only available for South Africa). The *ordistep* function from the 'vegan' package was used to select the most informative variables and build the 'optimal' model. Four separate RDAs were conducted with minor allele frequency as the response. Predictor variables in the first RDA were transformed geographic distances, and in the second were environmental distances. Lastly, two partial RDAs were performed, partitioning out the effect of transformed geographic distance and environmental variation from the total variation respectively. The *anova* function of the package 'vegan' was performed with 999 permutations (*Legendre, Oksanen & Ter Braak, 2011*) to test the significance of RDAs.

## RESULTS

### Sequencing and mapping

54,982,056 paired reads were obtained, with paired reads from each sampling site ranging from 1,368,372 to 7,429,328 (Table 3). After filtering reads for quality and adapters, and subsampling to a median, 7,432,397 reads, ranging from 222,741 to 750,736 per site, were aligned to the *Z. marina* reference genome (Table 3). The number of filtered subsampled mapped reads had no correlation with the number of SNPs ($r = 0.17$; $p > 0.05$) or outlier loci ($r = -0.05$; $p > 0.05$) identified.

**Table 2** Environmental variables included in BayeScEnv and IBE analyses.

| Environmental variable | Source |
| --- | --- |
| Macrophyte species measures | |
| Submerged macrophyte area (ha) | |
| Number of habitat types | *Adams, Veldkornet & Tabot (2016)* |
| Submerged macrophyte species richness | |
| the CLiMond dataset | |
| Annual mean temperature (°C) (Bio1)[a] | |
| Max temperature of warmest week (°C) (Bio5)[a] | |
| Min temperature of coldest week (°C) (Bio6) | |
| Annual precipitation (mm) (Bio12)[a] | *Kriticos et al. (2012)* |
| Precipitation of wettest quarter (mm) (Bio16)[a] | |
| Precipitation of driest quarter (mm) (Bio17) | |
| Annual mean radiation (W m-2) (Bio20) | |
| Annual mean moisture index (Bio28)[a] | |
| World Ocean Atlas | |
| Salinity (PSS) | *Zweng et al. (2013)* |
| Dissolved Oxygen (ml/l)[a] | *Garcia et al. (2013)* |
| Sea Surface Temperature (°C)[a] | *Locarnini et al. (2013)* |

**Notes.**
[a] indicates variables selected by the RDA as important contributors.

## Neutral and outlier loci

The complete simulated dataset consisted of 308 loci (Fig. 3). From this dataset, 101 outlier loci were detected by Lositan, while BayeScan and BayeScEnv detected 25 and five outlier loci respectively. The five outlier loci identified by the ecological association approach in BayeScEnv were correlated with precipitation of the driest quarter and annual mean moisture levels. By analysing allele frequencies of the non-simulated dataset, PCAdapt identified 38 outlier loci. All 169 outlier loci were removed from the complete dataset in order form a neutral-only dataset with which to examine patterns of neutral variation.

## Genome-wide variation

The number of SNPs identified by PoPoolation2 in the neutral and complete datasets ranged from 845 to 1,683 and 913 to 1,784 per sampling site respectively. The number of private SNPs per site was generally low and five populations did not have any private SNPs (Tables 3 and 4). The genome-wide average nucleotide diversity (Tajima's $\pi$) of the neutral and complete datasets ranged from 0.023 to 0.041 and 0.023 to 0.035, respectively. Allelic richness did not vary much between sites, ranging from 1.23 to 1.36. $\theta_W$ of the neutral and complete datasets were the same, ranging from 0.029 to 0.043 (Tables 3 and 4). The west and south coast sites, with the exception of Oostewaal (L2), exhibited marginally higher nucleotide diversity and $\theta_W$ than the east coast sites. Tests for deviations from neutrality produced genome-wide average Tajima's D that were negative for all sampling sites and ranged from $-0.723$ to $-0.275$ and $-0.706$ to $-0.273$, for the complete and neutral datasets respectively. Genetic diversity metrics calculated from the simulated datasets included expected heterozygosity (0.04 to 0.06) within each sampling site (Tables 3 and 4),

Phair et al. (2019), *PeerJ*, DOI 10.7717/peerj.6806

**Table 3  Summary statistics of RAD data and estimates of genetic diversity metrics per sampling site (refer to Table 1 for full names of abbreviations) for neutral dataset.**

| Sampling site | Raw reads | Mapped reads | Subsampled mapped reads | SNPs | Private SNPs | $\pi$ | $\theta$ | D | He | Ho | $F_{IS}$ |
|---|---|---|---|---|---|---|---|---|---|---|---|
| O | 5,862,886 | 1,457,363 | 743,255 | 1,278 | 2 | 0.034 | 0.041 | −0.714 | 0.04 | 0 | 1 |
| B | 4,314,436 | 1,114,902 | 746,984 | 1,683 | 3 | 0.035 | 0.042 | −0.722 | 0.04 | 0 | 1 |
| L1 | 4,997,550 | 1,153,894 | 750,031 | 1,473 | 2 | 0.034 | 0.041 | −0.698 | 0.04 | 0 | 1 |
| L2 | 1,368,372 | 222,741 | 222,741 | 1,027 | 0 | 0.025 | 0.031 | −0.616 | 0.06 | 0 | 1 |
| BR | 3,105,804 | 508,608 | 508,608 | 1,624 | 1 | 0.034 | 0.041 | −0.705 | 0.05 | 0 | 1 |
| K | 5,943,674 | 1,251,227 | 750,736 | 1,342 | 1 | 0.035 | 0.041 | −0.673 | 0.04 | 0 | 1 |
| SK | 5,882,100 | 1,360,205 | 748,113 | 1,387 | 0 | 0.035 | 0.042 | −0.675 | 0.04 | 0 | 1 |
| N | 4,296,798 | 568,703 | 568,703 | 845 | 0 | 0.028 | 0.034 | −0.654 | 0.05 | 0 | 1 |
| M | 3,991,420 | 475,470 | 475,470 | 914 | 1 | 0.025 | 0.032 | −0.637 | 0.05 | 0 | 1 |
| RB | 7,429,328 | 781,740 | 750,470 | 1,105 | 0 | 0.022 | 0.028 | −0.646 | 0.04 | 0 | 1 |
| MOZ | 4,136,268 | 719,319 | 719,319 | 598 | 0 | 0.026 | 0.028 | −0.276 | 0.05 | 0 | 1 |
| KEN | 3,653,420 | 447,966 | 447,966 | 1,480 | 6 | 0.029 | 0.043 | −0.324 | 0.04 | 0 | 1 |
| Total | 54,982,056 | 10,062,138 | 7,432,397 | – | 16 | – | – | – | – | – | – |
| Range | 1,368,372–7,429,328 | 222,741–1,457,363 | 222,741–750,736 | 845–1,683 | 0–6 | 0.023–0.035 | 0.029–0.043 | (−0.723)–(−0.275) | 0.04–0.06 | 0 | 1 |

**Notes.**

$\pi$, Tajima's $\pi$; $\theta$, Watterson's $\theta$; D, Tajima's D; He, Average expected heterozygosity; Ho, Average observed heterozygosity.

**Table 4  Estimates of genetic diversity metrics per sampling site (refer to Table 1 for full names of abbreviations) for complete dataset.**

| Sampling site | Number of SNPs | Number private SNPs | $\pi$ | $\theta$ | D | He | Ho | $F_{IS}$ |
|---|---|---|---|---|---|---|---|---|
| O | 1,362 | 2 | 0.034 | 0.041 | −0.716 | 0.04 | 0 | 1 |
| B | 1,784 | 3 | 0.035 | 0.043 | −0.723 | 0.04 | 0 | 1 |
| L1 | 1,577 | 2 | 0.034 | 0.041 | −0.700 | 0.04 | 0 | 1 |
| L2 | 1,091 | 0 | 0.025 | 0.031 | −0.616 | 0.06 | 0 | 1 |
| BR | 1,726 | 1 | 0.034 | 0.041 | −0.706 | 0.05 | 0 | 1 |
| K | 1,436 | 1 | 0.035 | 0.042 | −0.674 | 0.04 | 0 | 1 |
| SK | 1,483 | 0 | 0.035 | 0.042 | −0.676 | 0.04 | 0 | 1 |
| N | 913 | 0 | 0.028 | 0.034 | −0.651 | 0.05 | 0 | 1 |
| M | 997 | 1 | 0.026 | 0.033 | −0.636 | 0.05 | 0 | 1 |
| RB | 1,192 | 0 | 0.023 | 0.028 | −0.646 | 0.04 | 0 | 1 |
| moz | 668 | 0 | 0.027 | 0.029 | −0.273 | 0.05 | 0 | 1 |
| ken | 1,580 | 6 | 0.029 | 0.043 | −0.323 | 0.04 | 0 | 1 |
| Total | – | 16 | – | – | – | – | – | – |
| range | 913–1,784 | 0–6 | 0.023–0.041 | 0.029–0.043 | (−0.706) - (−0.273) | 0.04–0.06 | 0 | 1 |

**Notes.**

$\pi$, Tajima's $\pi$; $\theta$, Watterson's $\theta$; D, Tajima's D; He, Average expected heterozygosity; Ho, Average observed heterozygosity.

and the inbreeding coefficient, $F_{IS}$, which was uniform across sampling sites and equal to 1, for both the complete simulated dataset and the neutral simulated dataset. Although an $F_{IS}$ of 1 is partly due to the nature of the simulated data, generating multiple individuals from a highly clonal pool, it nonetheless indicates extremely high levels of inbreeding.

## Genome-wide differentiation and clustering

$F_{ST}$ values were estimated from the complete non-simulated dataset for pairwise comparisons of sites (Tables S1 , S2), with Fisher's exact tests showing no significant differentiation between pairs of sites for either dataset. Similarly, clustering analysis conducted in BAPS on neutral loci revealed no structure across sites, with all sites falling into one cluster ($K = 1$; $p < 0.05$; Fig. S1). Although there is no significant population structuring, the PCoA (Fig. 2) of pairwise $F_{ST}$ values for neutral loci suggests that the west and south coast sites, (except for Oostewaal-L2), are more closely related than the east coast sites. The same pattern was observed for the PCoA generated without Kenya (Fig. 2).

However, when the clustering analysis in BAPS included outlier loci, two clusters were detected ($p < 0.05$; labelled cluster one and two), with cluster one comprising samples from the west and south coasts, and cluster two including populations from the east coast of South Africa in addition to Mozambique and Kenya (Fig. 1B). Notably, one west coast site in Langebaan, Oostewaal (L2), groups with cluster two rather than cluster one (Fig. 1B).

PCoAs of pairwise $F_{ST}$ comparisons from the complete dataset and all outlier loci resulted in a similar, but slighter denser pattern than observed for the neutral dataset (Fig. 2). Sites from cluster one formed a tight group, relatively separate from the remaining sites. Sites from cluster two did not group as closely as those from cluster one, with Mozambique most differentiated. Moreover, Mozambique, followed by Kenya, exhibited much higher outlier allele frequencies than other sites (Table S3).

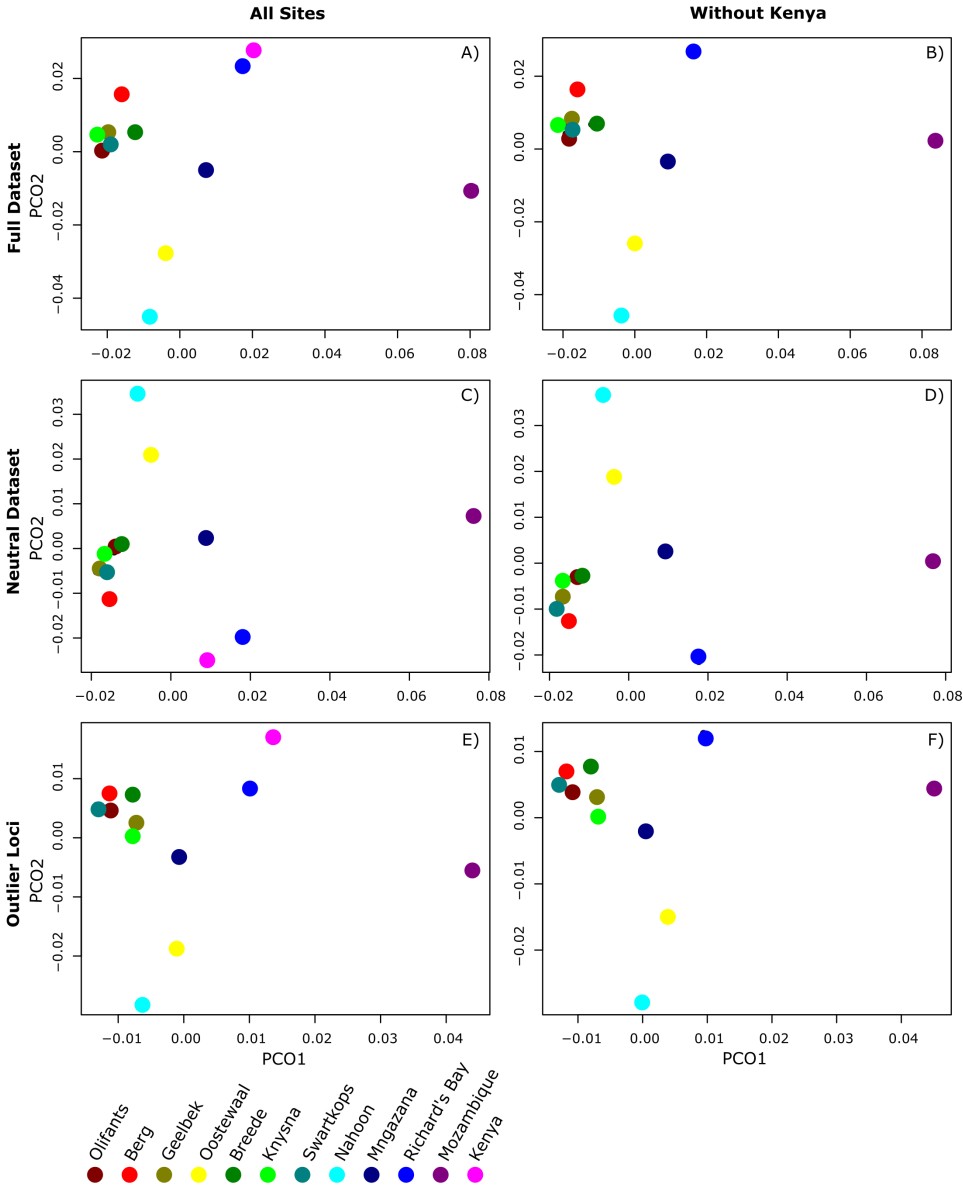

**Figure 2   Principle Coordinate Analysis (PCoA) plots of average pairwise.** Principle Coordinate Analysis (PCoA) plots of average pairwise $F_{ST}$ comparisons among all 12 sampling sites (A, C, E) and all sites excluding Kenya (B, D, F) for all loci in the complete dataset (A, B), the subset of loci contained in the simulated neutral dataset (C, D), and outlier loci (E, F). Sites grouping with cluster one and two are indicated by the red and green bar in the legend, respectively.

While some outlier loci were identified by more than one method, there was little overlap between outlier loci identified using the four different approaches (Fig. 3), with only three outliers shared between all four methods. However, irrespective of how many outlier loci are included, the frequency at which outlier loci occurred at each site reflects the two clusters identified using BAPS (Fig. 1B). ANOVA and post hoc TukeyHSD tests revealed

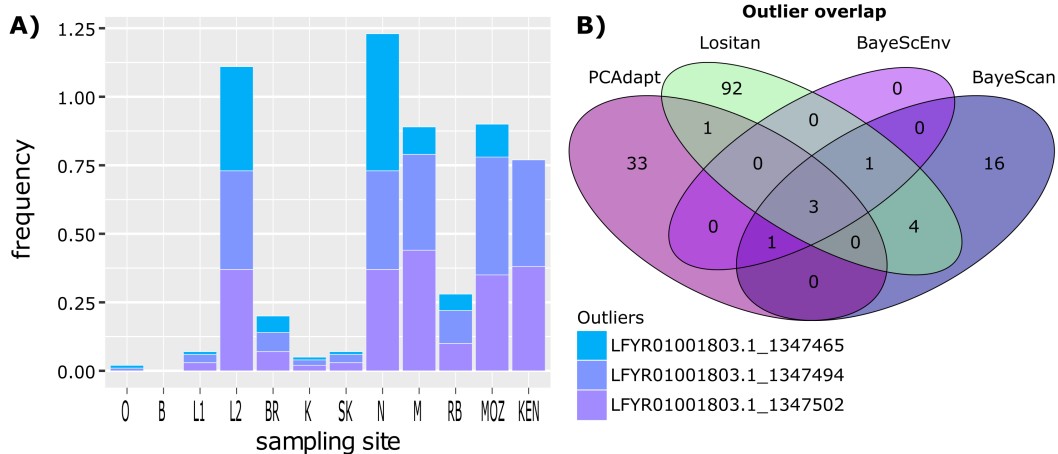

**Figure 3 Outlier frequency and identification overlap.** (A) The frequency of the three outlier loci identified by all four approaches (Lositan, BayeScan, BayeScEnv and PCAdapt) across sampling site. (B) Venn diagram illustrating the overlap between outlier loci identified using the four different approaches.

that outlier frequencies were not significantly different across sampling sites ($p > 0.05$) but that they were significantly different across the two clusters, with higher frequencies observed in cluster 2 than cluster 1 ($F_{1,24} = 66.61$, $p < 0.001$; Fig. 3). No private outliers were identified as all outlier loci occurred at two or more sites and all candidate outliers (identified by more than one method) occurred at most of the sites (Table S3).

## Functional annotation of candidate outlier loci

Two-thousand base pairs surrounding each of 10 candidate outlier loci were subjected to the Blast2Go pipeline. Although all of the 10 candidate outliers yielded significant hits when BLAST searches were conducted against the general NCBI database, Zosteraceae, *Z. marina* and *Z. muelleri*, the majority of these hits did not fall within gene regions of known function. GO terms (GO:0016020-IEA 'membrane' and GO:0016021-IEA 'integral component of membrane') were assigned to five of the 10 candidate outlier loci with BLAST matches to hypothetical and predicted proteins (Table S3).

## Habitat suitability for *Z. capensis* in the LGM

Multiple models from each algorithm met the TSS > 0.55 and AUC > 0.8 criteria and were retained to produce ensembles. Ensemble models obtained the following average validations scores: TSS = 0.654, AUC = 0.904, sensitivity = 92.11, specificity = 73.29. Predicted distributions of suitable habitat, in terms of SST, differed between present-day and LGM conditions, in terms of geographic location, extent and probability of occurrence (Fig. S2). The highest probability of occurrence can be seen on the south coast (up to ~25° longitude) and west coast (up to ~18° latitude) for the present-day projection, and on the western-south coast (up to ~21° longitude) and west coast (up to ~18° latitude) for the LGM projection. Ensemble models project an 11.05% loss and a 10.79% gain of suitable habitat from the LGM to present-day, with a 26.1% range shift. These shifts are most evident in the loss of suitable habitat on the south and south-east coasts (~21–27°

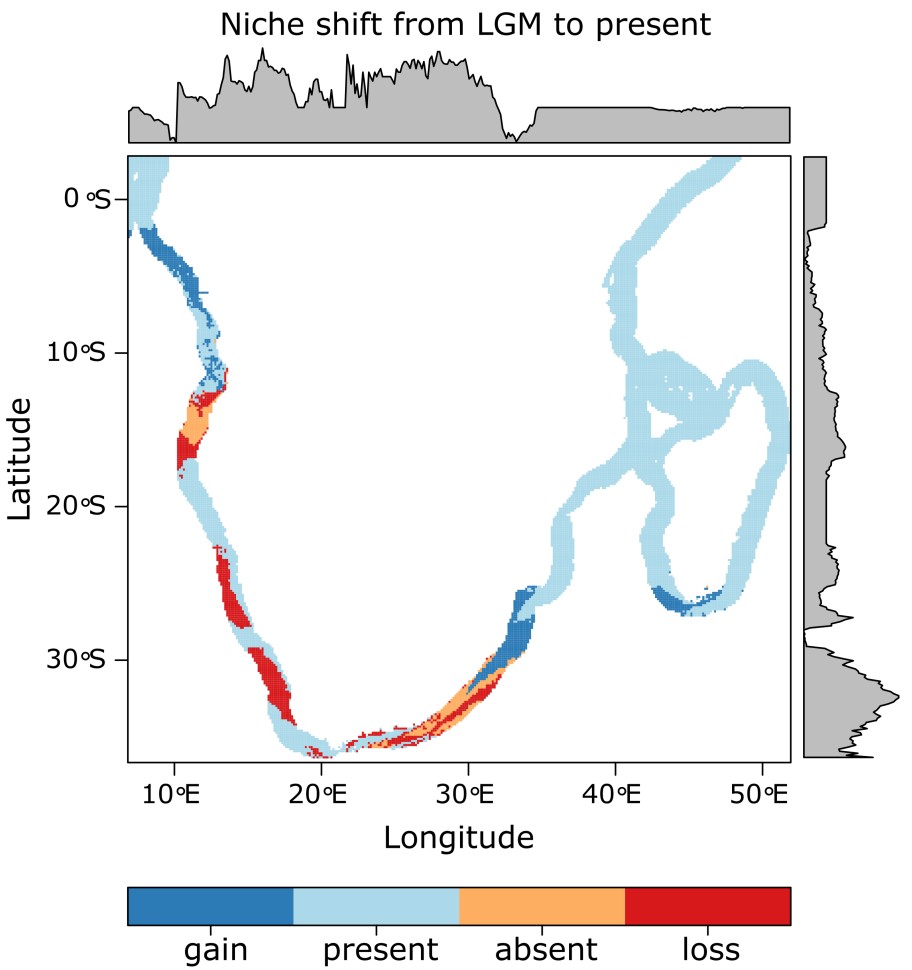

**Figure 4** **Projected changes in suitable habitat, in terms of SST, from the LGM to present with the probability of occurrence graphically represented along the *x* and *y* axes.** The western-south coast represents an area where suitable habitat, in terms of temperature, has remained stable from the LGM until the present day.

longitude), southern-west coast (∼30–35° latitude), and west coast (∼12–18° latitude), as well as the gain of suitable habitat on the northern-east coast of South Africa (∼30–35° latitude), the south coast of Madagascar and the northern-west coast of Africa (∼3–8° latitude; Fig. 4). Further, within a South African context, the western-south coast represents an area of stable temperature regime, where suitable habitat has occurred from at least as far back as the LGM until the present day (Fig. 4). This can also be seen in patches on the west coast and on the east coast of Africa (∼5–25° latitude).

## IBD vs IBE

Of the 11 environmental variables, seven were selected by the RDA as the most informative (Table 2). The pure RDA of genetic variation against transformed geographic distance was not significant ($P > 0.05$), but was significant when carried out against environmental variation, with 70.4% of the variation in the data explained by the retained environmental

variables. Unexpectedly, neither partial RDA analyses, the first conditioned on transformed geographic distance and the other on environmental variation, were significant. Although environmental variation explained such a high percentage of the variation observed in the data, partitioning out the effect of geographic distance on environmental variation rendered the association with genetic variation non-significant.

## DISCUSSION

### Genomic diversity of a threatened seagrass

Genomic variability did not differ greatly between populations with all sites displaying very low heterozygosity and a high inbreeding coefficient ($F_{IS} = 1$; Table 4). However, sites in cluster one did exhibit slightly higher levels of variability than those in cluster two (Table 4). With such high inbreeding coefficients, it is likely that this species does indeed rely heavily, if not solely, on clonal growth and vegetative reproduction, rather than sexual reproduction (Tables 3 and 4). In terms of reproductive strategy, clonality in seagrasses can vary between species with a continuum from monoclonality to meadows with high clonal diversity (*Van Dijk & Van Tussenbroek, 2010*), and the predominance of certain clonal lineages may indicate long-term selection on phenotypes. Such selection may be in response to environmental variables, where conditions are more favourable for clonal lineages, but may also represent shared ancestry prior to historic sea-level fluctuations reshaping the topography of the South African coastline (*Ramsay & Cooper, 2002*; *Compton, 2011*; *Ludt & Rocha, 2015*). In the case of *Z. capensis*, it is likely that a combination of historical (see below) and contemporary factors shape the patterns of observed genomic diversity. Importantly, *Z. capensis* is unlikely to be influenced by contemporary gene flow between its fragmented and isolated meadows, being restricted to sheltered and low wave action environments (*Van Niekerk & Turpie, 2012*). In addition, the lack of recorded sexual reproduction in this species through flowering (*McMillan, 1980*; D Pillay, pers. comm., 2014) is likely to contribute to maintaining clonal populations throughout the range, with important implications for potential restoration efforts in the region.

### Shared adaptive divergence across two genomic clusters shaped by historical dynamics

Although neutral variation can reveal much about a species demographic history, in many cases, patterns revealed from outlier loci can provide unique insights into evolutionary potential and patterns of resilience (*Stapley et al., 2010*; *Guo et al., 2015*; *Funk et al., 2016*; *Gaither et al., 2018*). Particularly in marine systems where gene flow is generally presumed to be high, signals of outlier loci can help detect population structure (*André et al., 2011*; *Freamo et al., 2011*; *Hess et al., 2013*; *Candy et al., 2015*; *Araneda et al., 2016*; *Tigano & Friesen, 2016*; *Attard et al., 2018*). For example, Atlantic herring in the Baltic and North Sea (*André et al., 2011*), Atlantic salmon in eastern Canada (*Freamo et al., 2011*), and Chilean blue mussels (*Araneda et al., 2016*), all exhibit little to no structure in terms of neutral variation but reveal significant population structure for outliers putatively under selection. In *Z. capensis*, despite the generally low levels of genomic variation detected across its range, clustering analyses revealed differentiation of all populations into two major

clusters when outliers that may represent putatively adaptive variation were considered in addition to neutral data (Fig. 1B). Cluster one comprised of sites from the west and south coast, and cluster two sites from the east coast in addition to one west coast site at Langebaan, Oostewaal, also grouping with this cluster (Fig. 2). In addition, PCoAs support the clustering analyses, with both neutral and adaptive variation (Fig. 2). Interestingly, the PCoAs indicate that east coasts sites are more distantly related than the west and south coast sites. Therefore, east coast sites may have had an earlier origin which supports the likelihood of a refugial area for *Z. capensis* on the east coast.

Temperature-based ensemble models, however, suggest reduced, and more fragmented seagrass habitat along the South African south coast, that is likely to have divided *Z. capensis* into two clusters with the south-western and east coast possibly representing refugial areas during the LGM, with subsequent dispersal into its present-day distribution. A refugial area on the south-western coast may explain the presence of both clusters in Langebaan Lagoon. Notably, this split between the clusters roughly coincides with the split between described temperate and sub-tropical bioregions (Sink et al., 2012) along which phylogeographic breaks have been recorded for marine coastal species (*Von der Heyden, 2009*; *Teske et al., 2011*), including one other saltmarsh plant (*Potts, Veldkornet & Adams, 2016*). Given lack of gene flow and apparent high levels of clonality in *Z. capensis*, the structure detected through outlier loci most likely reflects ancestral adaptation during conditions more conducive to gene flow or incomplete lineage sorting during post-LGM expansion.

Interestingly, although our historical models were based on environmental variables, they broadly mirror the findings of changes in topology and composition of the South African coastline during the last 70,000 years (*Toms et al., 2014*). During the past 45,000 years, lowered sea level stands of up to −120 m caused significant shifts from rocky to sandy/muddy shorelines which isolated populations of obligate rocky shore species. Although *Z. capensis* is found in present-day estuaries, the latter would also have been affected, although the extent of change is unknown. Our findings, in combination with *Toms et al. (2014)*, do however show that combinations of abiotic changes have the potential to influence the population dynamics of marine and estuarine species in the Atlantic/Indian Ocean transition zone, as they have done in Australia over exceedingly short time scales (*Puritz et al., 2012*).

We provide evidence for a pattern of shared outlier loci for populations across a distinct environmental gradient and large geographic span, despite the presence of two genomic clusters that appear to represent distinct historical lineages. This is in contrast to our hypothesis of distinct population-level signals of adaptive variation as seen in previous studies (*Williams & Oleksiak, 2008*; *Perrier et al., 2013*; *Ravinet et al., 2016*). All outlier loci were shared among sites with no populations harbouring private outliers (Table S3), suggesting the same genomic basis for each *Z. capensis* meadow with the same suite of genes under selection across sites in response to the various environmental gradients. However, differences in the frequencies of these outlier loci across sites provides the foundation for the two clusters, with sites from cluster one exhibiting outlier loci at lower frequencies compared to cluster two (Fig. 3). Notably, this pattern of differential outlier allele frequencies could be observed even when only considering three outlier loci (Fig. 3).

The same clustering pattern is detected with both non-simulated and simulated datasets, which were necessary in order for the analysis of pooled data with certain software. This similarity demonstrates that simulated datasets can be used to detect biologically significant evolutionary patterns, regardless of the over-simplifications these simulated datasets may introduce, or the number of SNPs one chooses to employ.

There have been numerous other studies that also report high levels of shared adaptive variation across sites, such as that in Atlantic salmon of eastern Canada, where the allele frequencies of shared outlier loci were used in to assign individuals to their region of origin, assisting with stock management (*Freamo et al., 2011*). Similarly, in Pacific and Atlantic sticklebacks different allele frequencies of shared outlier loci have been used to distinguish marine and freshwater populations (*Jones et al., 2012*). At a smaller scale in western Canada, most outlier loci in sticklebacks were specific to single watershed regions (*Deagle et al., 2012*). Likewise, few shared outlier loci were observed in the periwinkle, *Littorina saxatilis*, in Sweden (*Ravinet et al., 2016*), suggesting that the shared or private nature of outlier loci might be highly context specific. Despite the potential for high levels of gene flow and similar selective pressures, *L. saxatilis* populations displayed a considerable amount of unshared genomic divergence, possibly due to complex polygenic traits involved in habitat adaptation.

## The contribution of IBD and IBE towards the spatial arrangement of genomic variability in *Z. capensis*

Despite the low probability of connectivity between sites, due to both the isolated nature of estuaries and the lack of sexual reproduction recorded for this species, geographic distance (IBD) was not a significant driver of the observed genomic variation, with some evidence for IBE in this system. However, because of spatial autocorrelation with environmental variables chosen in this study, there appears to be a large spatial component shaping genomic variation, which cannot be separated from the effect of environmental variables. Our results suggest that IBE plays an important role in shaping genomic variation in this seagrass, in particular dissolved oxygen, annual mean moisture, precipitation and temperature related environmental variables, that were all significantly associated with outlier loci. Although the functional relevance of these outliers is unclear, this may indicate some level of adaptive variation. In addition, it is likely that fine-scale environmental variation, specific to each of the estuaries from which *Z. capensis* were sampled additionally contributes to IBE. However, given the lack of *in-situ* measurements of important environmental variables for coastal South and eastern Africa, this is not possible to determine at this stage.

## CONCLUSIONS

*Zostera capensis* along the African coast have not been observed to reproduce sexually, and high clonality combined with low genomic diversity increases their vulnerability to direct human pressures and a changing climate. Genomic similarity between sites however may confer a level of resilience as meadows particularly in the context of restoration (*Hughes & Stachowicz, 2004*; *McKay et al., 2005*; *Reynolds, McGlathery & Waycott, 2012*; *Baums, 2008*).

Even though there is no significant structure based on neutral loci, the clustering of sites into west and east based on outlier loci may indicate different levels of selection on the same suite of genes, which is not implausible given the environmental and ecological gradients that characterise our study area. Shared outlier loci among independent lineages, coupled with differences in the frequencies of those outlier loci among populations correlated with environmental variability is consistent with the potential for local adaptation. Although we could not directly test for local adaptation, and recognise the need for reciprocal transplant experiments (*Kawecki & Ebert, 2004*), the frequency differences among outlier loci may indicate some functional variation, which could in turn influence how different populations respond to changing environmental conditions. Additionally, the two clusters appear in part to be shaped by historical environmental variation, with each cluster linked to climatically stable refugia on the south-western and east coasts. As such, it is important to protect and maintain distinct populations across the distributional range of keystone species in order to safeguard this seagrass and its vital ecosystem services into the future (*Hughes & Stachowicz, 2004*). In particular, while Langebaan Lagoon may serve as an ideal focal point for conservation on the west coast, areas on the east coast of South Africa, Mozambique and Kenya should also be targeted for conservation in order to increase resilience and reduce the risk of widespread loss.

## ACKNOWLEDGEMENTS

This study would not have been possible without the assistance of the following individuals in collecting seagrass samples from across southern Africa: Marcel van Zyl, Dr Jaco Barendse, Dr Kyle Smith, Prof. Janine Adams, Rob Nettleton, Dr Leon Vivier and Dr Nina Wambiji.

### Funding

This study was supported by the South African National Research Foundation (NRF) through a Scarce Skills Doctoral Scholarship to Nikki Leanne Phair, the Western Indian Ocean Marine Science Association (WIOMSA) through the MARG I grant to Sophie von der Heyden and NSF OA #1416889 to Robert John Toonen. The funders had no role in study design, data collection and analysis, decision to publish, or preparation of the manuscript.

### Grant Disclosures

The following grant information was disclosed by the authors:
South African National Research Foundation.
Western Indian Ocean Marine Science Association.
MARG I.
NSF OA #1416889.

# PeerJ

## Competing Interests

Robert Toonen is an Academic Editor for PeerJ.

## Author Contributions

- Nikki Leanne Phair conceived and designed the experiments, performed the experiments, analyzed the data, prepared figures and/or tables, authored or reviewed drafts of the paper, approved the final draft.
- Robert John Toonen contributed reagents/materials/analysis tools, authored or reviewed drafts of the paper, approved the final draft.
- Ingrid Knapp performed the experiments, contributed reagents/materials/analysis tools, authored or reviewed drafts of the paper, approved the final draft, sequencing library preparation.
- Sophie von der Heyden conceived and designed the experiments, contributed reagents/materials/analysis tools, authored or reviewed drafts of the paper, approved the final draft.

## Field Study Permissions

The following information was supplied relating to field study approvals (i.e., approving body and any reference numbers):

The collection of plant material was approved by SanParks and Cape Nature (permit number 0028-AAA008-00159), DAFF and DEA (permit number was RES2014/103).

## Data Availability

The raw data is available at National Center for Biotechnology Information's (NCBI's) Sequence Read Archive (SRA; PRJNA503110) and georeferenced at GeOMe at the project: *Zostera capensis* pooled RADseq. To access the data go to https://geome-db.org/query and search for "*Zostera capensis*" in the "any term" search box.

## Supplemental Information

Supplemental information for this article can be found online at http://dx.doi.org/10.7717/peerj.6806#supplemental-information.

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
