# Peer review of "Shared genomic outliers across two divergent population clusters of a highly threatened seagrass"

_PeerJ, doi:10.7717/peerj.6806_

## Round 0.1 · original submission · Minor Revisions

The manuscript is largely well-written and presents an interesting addition to seascape genetic diversity and adaptive potential. I agree that a number of sentences are rather long and cumbersome for the reader. I also recommend that particular attention is payed to clarifying details in the methods and discussion. Below are some additional comments of my own:

Line 20-23 – could this sentence be simplified or tightened up a little? It reads as being a little long and cumbersome.
Line 56 – RADseq should start a new paragraph
Line 61-63 – I disagree with this statement. Fewer studies have been conducted in natural settings than controlled settings, but really there are numerous studies assessing spatial variation with respect to outlier loci. (I myself have performed some, as have the reviewers). This needs to be toned down or made much more specific, if that is what is actually the case.
Line 71-73 – what exactly is meant by ‘the balance of IBD and IBE within marine systems is becoming more apparent’? Can you explain the ‘balance’?
Line 86 – authority and family for Zostera
Line 134 – Do you mean Inhaca in Mozambique and Shimoni in Kenya? At present, for people less familiar with the area it is unclear if Inhaca and Mozambique are separate locations or not. If it refers to Inhaca within Mozambique, can Mozambique and Kenya be placed in parentheses to make it clear that these are only two locations within the broader context of Mozambique and Kenya?
Methods – make it very clear what is happening with the samples in terms of pooling
Line 140 – perhaps change the end of this sentence to ‘locally extirpated’
Line 181 – were, not was
Line 191 – create, not creating
Line 211 – which ones exactly – be specific. It is not, in this case, acceptable to refer the reader to the supplementary for all details relating to the outlier detection considering it is such a big part of your study. Please include a brief (at least names and citations for software) description of what was done.
Methods – several notations commonly used in population genetics are not being italicised as they should be. Please remedy this (e.g. Fst, K).
Line 220 – Interproscan requires a citation/reference – the website details how to cite this software appropriately
Line 258-263 – this sentence is too long, please shorten
SDM models – why use only sea surface temperature and precipitation? Why not include information on salinity, O2 etc. considering some of these populations are estuarine? Some further detail or justification would be nice.
General – the R software needs a citation for the R Core Development Team.
Line 319-326 – all the outlier detection programmes need citations – although adding more in the methods should fix this problem
Results – perhaps I am missing something… how can the Ho and Fis be 0 and 1 respectively? There were clearly SNPs detected, so even as a population average this doesn’t seem to make sense. I think this section needs some additional explanation because it is not clear to me, unless they are 100% clones all the time, how this is possible. Yet, if they are ramets the SNP counts should not be so high unless it is considered sequencing error. I appreciate they likely do have a lot of selfing. Perhaps remind the reader (especially in the discussion) why this is a ‘good’ result.

Figure 1 – please provide some country/state labels for the map to provide context for the reader.
Table 1 – Can you provide some more detail regarding the sampling strategy and number of samples per site. In the table there are typically 30 samples per site, but how does this break down over the 5 seagrass beds and two distant sites for each location? Is this 2 sub-sites per location with 5 beds each for 3 samples per sub-site per bed per location? Please clarify this a bit more in the text (lines 134-136).

Reviewer 1 ·

Basic reporting

1. Not all citations, e.g. Knapp et al. 2016, are in the references this needs to be double checked and addressed. Also, the reference for Widmer and Lexer 2001 is not cited in the paper. As a result, I couldn’t easily check on the method by Knapp et al. 2016 - presumably one of the authors?
2. Lines 13-15 “However, how patterns of natural variation and population genomic structure are shaped by adaptive processes remains unknown.” and the statement on lines 61 – 63 about studying spatial variation of outliers needs to be toned down quite a bit or reworded because the likely intended meaning did not come through. This is very close to what the entire field of landscape and ecological genomics are attempting to do. There are a lot of papers on using SNP data and environmental variables to look at landscape level variation, adaptation, and even susceptibility to disease. For example, the R package LEA and the lfmm function within LEA (https://doi.org/10.1111/2041-210X.12382) or studies of environmental association. Indeed, the RDA, BayeScan, BayeScEnv, and PCAdapt analysis that the authors use has been used similarly in other plant species. Perhaps the authors meant that it hasn’t been done frequently in pooled ezRAD studies or studies of seagrasses and the like?
3. I would like to see some form of Fig. S2 in the paper instead of the supplements. It is an important enough visualization of the data that it should be part of the main body.
4. Line 169, please provide a URL for the GeOMe website.
5. On line 56 when the authors start talking about RAD-seq as a technique to gather genetic data the flow of the introduction might be improved by adding a paragraph here. The beginning of the introduction is very dense with a lot of long, run-on sentences that can fatigue the reader. The reasoning and flow are good, but readability would be improved by shortening some of the sentences.

Experimental design

1. Within the methods I would like to know which samples were pooled? It isn’t clear until the results when Table 3 is presented that the samples were likely pooled by location. This needs to be more transparent in the methods.
2. I would like to see a contingency table analysis done and a table added to the supplements for the outlier loci to statistically support the claim that the frequency of outliers is different between the two groups (method: Slatkin M. 1995. A measure of population subdivision based on microsatellite allele frequencies. Genetics. 139: 457–462; recent application of: Gaither et al. 2018. Genomics of habitat choice and adaptive evolution in a deep-sea fish. Nature Ecol & Evol 2(4): 680.)
3. I can’t evaluate the method used to generate the values in supplementary Table S3 because the script was not provided. Scripts really should be submitted to a data archive such as GitHub or whatever is the authors favorite repository to ensure that experiments, including the bioinformatics, can be duplicated.

Validity of the findings

The study was well designed and the interpretation of results was clear and accurate.

Additional comments

Summary: The authors are providing valuable data about the genomic variation within a vulnerable, clonal seagrass species suffering from population declines and habitat loss/degradation. The authors are characterizing the standing genetic variation and adaptation potential of the species and have hypothesized that environment and distance will be equally important drivers of population differentiation and diversity. They collected a reasonable number of samples and cites. The use of pooled data seems to save on upfront costs of adapters/indexes necessary for NGS and is appropriate for the question being addressed. They found two distinct “population” clusters generated by differences in outlier frequency rather than private alleles. The authors did find support, although not strong, for their hypothesis that IBD and IBE are equally important in structuring the population. The management implications (naming two areas to concentrate on for preservation) and future directions (i.e. common garden experiments) were clear and well said.

General note: The number of compound and run-on sentences, especially in the introduction, reduces the readability of the paper. I suggest going through and shortening some sentences to improve this.

·

Basic reporting

This is an extremely well written paper, well referenced and has a big message for seagrass researchers. The lack of genomic variation across southern Africa is an unexpected outcome as is the lack of IBD signal.

The pooling of individuals at the location scale concerned me and the effect of pooling needed to be discussed a little in the discussion. I was very confused about how Fis can be calculated from pooled samples - this is probably my ignorance. Clearly Fis values of 1 indicate extreme levels of inbreeding. Is this a product of the low sample size combined with the pooling of these samples?

Clearly by reducing individual variation you have reduced contemporary signatures of genomic variation. Can you discuss this further?

Experimental design

The design of collections was confused - I was not sure about whether the 3 sites at each location were all pooled or whether only samples from sites were pooled. Overall, it is hard to reconstruct the sampling design. I recommend expanding this. The table indicates most locations have 30 samples - no details of sampling distances?

Validity of the findings

Other than my own confusion about the design and methods above - this is a robust analysis and should be published
I would like some more discussion on outlier SNPs separating out populations on west and east coasts of southern Africa but also more details of the isolation by environment patterns seem among estuaries.

Additional comments

I commend the authors on an excellent and careful analysis and a clearly written manuscript

---

## Round 0.2 · accepted · Accept

The manuscript has been much improved as a result of the changes and attention to detail. During production, please take care to ensure that all Fis references are italicized also (it seemed unnecessary to go through a round of minor revisions for this).

#